# Hydrochemical Characteristics of Earthquake-Related Thermal Springs along the Weixi–Qiaohou Fault, Southeast Tibet Plateau

**Huiling Zhou** [1], **Xiaocheng Zhou** [2,*], **Hejun Su** [1], **Ying Li** [2], **Fengli Liu** [2], **Shupei Ouyang** [2], **Yucong Yan** [2]
**and Ronglong Bai** [3]

[1] Gansu Lanzhou Geophysics National Observation and Research Station, East Mountains West Road 450, Lanzhou 730000, China; 15609413188@163.com (H.Z.); suhejun@126.com (H.S.)
[2] United Laboratory of High-Pressure Physics and Earthquake Science, Institute of Earthquake Forecasting, CEA, Beijing 100036, China; subduction6@hotmail.com (Y.L.); liufengli9723@163.com (F.L.); ouyangshupei888@163.com (S.O.); yanyucong2020@163.com (Y.Y.)
[3] College of Resources and Environmental Engineering, Tianshui Normal University, Tianshui 741001, China; baironglong@126.com
\* Correspondence: author: zhouxiaocheng188@163.com

**Abstract:** The Weixi–Qiaohou Fault (WQF) is considered an important zone of the western boundary of the Sichuan–Yunnan block, and its seismicity has attracted much attention after a series of moderate–strong earthquakes, especially the Yangbi $Ms$6.4 earthquake that occurred on 21 May 2021. In the present research, we investigate major and trace elements, as well as hydrogen and oxygen isotopes, of 10 hot springs sites located along the WQF, which are recharged by infiltrated precipitation from 1.9 to 3.1 km. The hydrochemical types of most analyzed geothermal waters are $HCO_3SO_4$-Na, $SO_4Cl$-NaCa, and $SO_4$-Ca, proving that they are composed of immature water and thus are characterized by weak water–rock reactions. The heat storage temperature range was from 44.1 °C to 101.1 °C; the circulation depth was estimated to range between 1.4 and 4.3 km. The results of annual data analysis showed that $Na^+$, $Cl^-$, and $SO_4^{2-}$ in hot springs decreased by 11.20% to 23.80% north of the Yangbi $Ms$5.1 earthquake, which occurred on 27 March 2017, but increased by 5.0% to 28.45% to the south; this might be correlated with the difference in seismicity within the fault zone. The results of continuous measurements of NJ (H1) and EYXX (H2) showed irregular variation anomalies 20 days before the Yangbi $Ms$6.4 earthquake. In addition, $Cl^-$ concentration is more sensitive to near-field seismicity with respect to $Na^+$ and $SO_4^{2-}$. We finally obtained a conceptual model on the origin of groundwater and the hydrogeochemical cycling process in the WQF. The results suggest that anomalies in the water chemistry of hot spring water can be used as a valid indicator of earthquake precursors.

**Keywords:** thermal spring; isotopes; hydrogeochemistry; earthquake

## 1. Introduction

The Sichuan–Yunnan rhombic block represents the most active block-lateral extrusion in the eastern margin of the Qinghai–Tibet Plateau, where seismic activity is intense and frequent [1–5]. The Weixi–Qiaohou fault (WQF) is located at its western edge, and it connects with the Honghe Fault to the south and the Jinshajiang fault to the north. Since historical earthquakes along the WQF are not significant, seismologists have paid little attention to its activity for a long time. Nevertheless, in recent years, there have been several earthquakes along the WQF, such as the 2013 Eryuan $Ms$5.5 and $Ms$5.0 earthquakes, as well as the 2017 Yangbi $Ms$5.1 earthquakes; in particular, the $Ms$6.4 occurred on 21 May 2021, which may indicate that seismicity in the western margin of the Sichuan–Yunnan block has been enhanced. Therefore, the future seismic activity, and related risk, of the WQF should be closely related. As the product of physical and chemical changes in the earth's interior, in-depth study of the physicochemical parameters and isotopic composition characteristics of the deep fluid may be able to extract useful information related to earthquake precursors.

There have been numerous studied related to the hydrological changes in groundwater chemistry before and after earthquakes. These include variations in hydrochemical composition [6–12], concentrations of dissolved ions [13–26], and stable isotope ratios [27–33]. Many geochemical anomalies generated prior to seismic events are usually caused by deep fluid pressure changes resulting from crustal deformation. Most researchers believe that changes in groundwater chemistry are closely related to seismic events, but the understanding of the parameters of earthquake precursors is still controversial in earthquake prediction studies. Sufficient and accurate observation hydrogeochemical data are critical for studying hydrological processes during earthquakes. Skelton et al. (2014) and Skelton et al. (2019) used long-term observed data and reported changes in $\delta^2H$ values and Na concentrations in Ice Age groundwater before two consecutive earthquakes [23,24]. Petrini et al. reported geochemical anomalies in some groundwaters from the Udine province due to local seismic events and associated crustal deformation processes [34].

In this study, the hydrogeochemistry of 10 natural geothermal hot springs exposed along the WQF was investigated by measuring the stable isotopes of hydrogen ($\delta^2H$) and oxygen ($\delta^{18}O$), as well as the concentrations of dissolved major and trace elements. The latest samples of groundwater were collected in the three days following the 2021 Yangbi *Ms*6.4 earthquake. Continuous measurements were performed every three days in two hot springs, including NJ hot spring (H1) since December 2018 and EYXX hot spring (H2) since March 2021. In more detail, we studied the hydrogeochemical temporal and spatial variation characteristics of hot spring water in the WQF, and we also investigated the hydrogeochemical anomalies of hot springs associated with the 2021 Yangbi *Ms*6.4 earthquake and the 2017 *Ms*5.1 earthquake.

## 2. Geological Setting

Since the early Cenozoic era, the eastward extrusion of the Qinghai Tibet Plateau has led to the formation of Sichuan–Yunnan rhombic blocks, and since the early Cenozoic era, eastward compression of the Tibetan Plateau has led to the formation of the Sichuan–Yunnan rhombic block [35–37]. The study area is located in the hinterland of the Sichuan–Yunnan block, located in the western Sichuan Province, China; this is the most active block, with the strongest lateral extrusion, on the eastern edge of the Qinghai Tibet Plateau, and it experiences frequent earthquakes (Figure 1). Blocked by the South China block, it moved to the southeast and rotated clockwise around the East Himalayan tectonic junction [38–41]. The WQF starts from Baijixun, at the eastern foot of the Xuelong mountain in the north, passes through Weixi, Tongdian, Qiaohou, and Pingpo, and then spreads along the western edge of the Weishan basin, with a total length of about 280 km. The WQF fault can be divided into three segments: the north, middle, and south segments. During the neotectonic movement, the former two were mainly characterized by dextral strike–slip movement: the strike has an NW direction, the dip is to the SW or NE, while dip angles range from 60° to 80° [42]. At the eastern boundary of the Lamping–Simao depression, Mesozoic sedimentary layers are widely developed on its western side, while the eastern side is mainly the Cang Shan metamorphic zone [43–46]. Moreover, the WQF is connected to the Jinsha River fault to the north and the Red River fault to the south, forming the western boundary of the Sichuan–Yunnan rhombic block. The Yangbi *Ms*6.4 earthquake occurred in the middle segment of the WQF, which has exhibited dextral strike–slip and normal faulting activity, with an average horizontal slip rate of 1.25 mm/year since Quaternary times [45,46]. From the perspective of regional geological evolution and neotectonic movement history, both the WQF and Red River faults have a common geological evolution history and structural deformation mechanism, which has changed from left-handed strike-slip movement in early Cenozoic times to right-handed strike-slip movement in Pliocene; it falls on a continuous straight line with the Red River fault zone. It is the northern extension of the Red River fault zone and plays an important role in the formation, evolution, and migration of the Sichuan–Yunnan active block. Therefore, it is undoubtedly of great theoretical and practical significance to strengthen the study of WQF

activities for an in-depth understanding of the seismic geological background and tectonic deformation mechanism in northwest Yunnan.

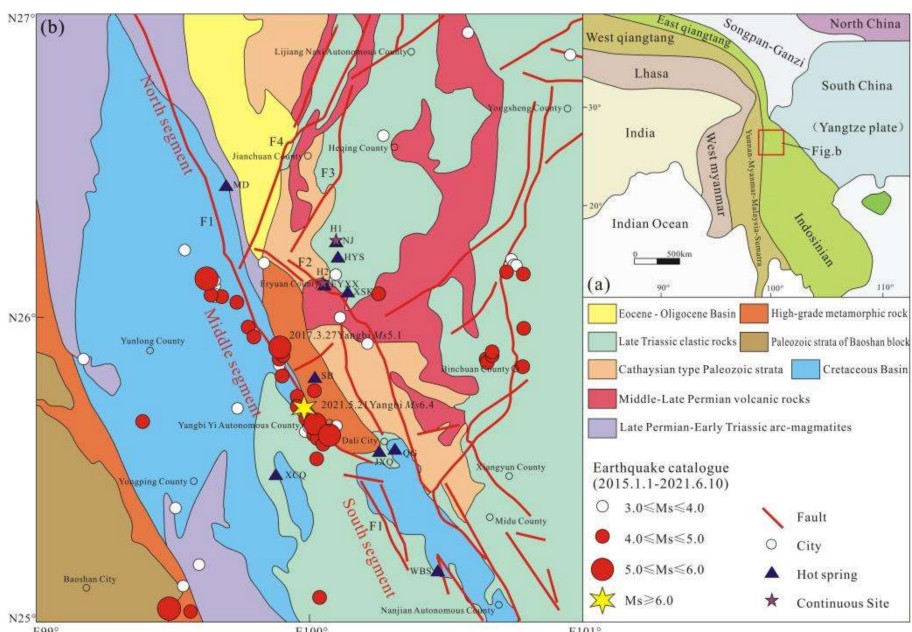

**Figure 1.** (**a**) location of the area of this study; (**b**) geological map in WQF and sampling positions (F1: Weixi–Weishan fault; F2: Red River fault; F3: Heqing–Eryuan fault; F4: Lijiang–Jianchuan fault).

## 3. Data and Methods

Groundwater samples were collected multiple times from 10 hot spring locations (MD, NJ, EYXX, HYS, XSK, SB, QG, JXQ, WBS, XQC, ID: W1, W2, W3, W4, W5, W6, W7, W8, W9, W10) along the WQF in this study (Figure 1b). Tectonically, all of the hot springs observed are distributed along the WQF and interrelated fault zones (Figure 1b). W1 is located in the north segment, W6, W7, W8, and W10 are located in the middle segment and W9 is located in south segment of WQF. W2, W3, W4, and W5 are not located in the adjacent Red River fault zone and the intersection of the Red River fault with Heqing–Eryuan fault. We measured stable isotope ratios for hydrogen ($\delta^2H$) and oxygen ($\delta^{18}O$) and the concentration of dissolved major ions ($K^+$, $Na^+$, $Mg^{2+}$, and $Ca^{2+}$) and anions ($Cl^-$, $SO_4^{2-}$, $CO_3^{2-}$, $HCO_3^-$) and trace elements. The water temperatures were measured in the field using a digital thermometer with an error of 1%. Water samples for stable isotope ($\delta^2H$ and $\delta^{18}O$) analysis were preserved in polyethylene bottles with no chemical agents added. They were measured with a liquid water isotope analyzer LWIA 912–0008 from Los Gatos Research at the Key Laboratory of Crustal Dynamics, China Earthquake Administration. Three certified reference materials (CR*Ms*), No. GBW04458, GBW04459, and GBW04460, from the Institute of Hydrogeology and Environmental Geology, Chinese Academy of Geological Sciences, were used to calibrate the isotopes. The sample results are presented as δ values relative to the Vienna Standard Mean Ocean Water in units of ‰. The precision was 0.30‰ for $\delta^2H$ and 0.08‰ for $\delta^{18}O$, calculated from the mean and standard deviation (1σ) of all measured samples. The instrument accuracy was characterized by comparing the measurement of CR*Ms* between different instruments. The concentrations of cations ($K^+$, $Na^+$, $Mg^{2+}$, and $Ca^{2+}$) and anions ($F^-$, $Cl^-$, $Br^-$, $NO_3^-$, and $SO_4^{2-}$) in the water samples were measured using a Dionex ICS 900 ion chromatography system in the Seismic Fluid Laboratory of the Institute of Earthquake Science, China Earthquake Administration, with the precisions within 2%. The concentrations of $CO_3^{2-}$ and $HCO_3^-$ were measured via standard titration procedures with a ZDJ 100 potentiometric titrator (precisions within ±2%). For the chromatography calibration, standard samples were measured before and after measuring each water sample batch. The data were evaluated by the ion balance (ib),

which can be calculated according to Equation (1) [18]. Trace elements were analyzed at the Test Center of the Research Institute of Uranium Geology by an element inductively coupled plasma source mass spectrometer ICP-MS [47]. Physicochemical parameters and the analytical data of major elements and trace elements of the spring waters are shown in Supplementary Materials Tables S1 and S2.

$$\text{ib}(\%) = 100 \times \frac{\sum cations - \sum anions}{\sum cations + \sum anions} \tag{1}$$

## 4. Results and Discussion

### 4.1. Stable Isotope Compositions and Recharge Sources of Hot Springs

Values of $\delta^2$H and $\delta^{18}$O varied from $-118.6$‰ to $-82.8$‰ and from $-15.2$‰ to $-11.3$‰, respectively, three days after the 2021 Yangbi $Ms$6.4 earthquake (Table 1). The global (GMWL) and regional atmospheric precipitation line (LMWL) are the most commonly used concepts. The LMWL equation for the Tibetan Plateau is given by the equation $\delta^2$H = 8.4 $\delta^{18}$O + 16.72 [48]. The isotope composition of the 10 hot springs was distributed around GMWL and LMWL before 2021, providing significant information that they were recharged by infiltrated precipitations. The recharging elevation ($\delta D = -0.03\,H-27$) [40] has been calculated to be about 1.9–3.1 km. However, the latest results show that it shifted to the right (Figure 2). Furthermore, an interesting characteristic of stable $\delta^{18}$O and $\delta^2$H is that the isotope composition is more negative than previous results. From 2017 to 2021, the decreasing ranges of $\delta^2$H at W2, W4, W5, W6, W7, W8, and W9 are $-19.1$‰, $-22.4$‰, $-17.4$‰, $-22.2$‰, $-18.5$‰, $-17.3$‰, and $-21.61$‰, respectively. Additionally, at W1, in the northern part of the fault, $\delta^2$H corresponds to 5.1‰, from 2018 to 2021. Furthermore, $\delta^{18}$O had a significant negative change by $-1.1$‰ in W3, from 2018 to 2021, while W5, W6, W7, and W8 changed by $-0.8$‰, $-1$‰, $-1.7$‰, and $-2.2$‰, from 2017 to 2021, respectively. Ranges of W1 in the northern segment and of W9 in the southern segment correspond to $-0.4$‰ and $-0.3$‰, respectively, which is less than the oxygen isotope variation of the middle segment. Generally, groundwater isotopic evolution has three important sectors, namely evaporation, exchange and mixing. In the equilibrium state, the evaporation and exchange are very slow [49]. A previous four consecutive years of study showed small isotopic variation in $\delta^2$H (2.49‰) and $\delta^{18}$O (0.52‰) from an 82 °C hot spring in Yunnan Province, China [50]. The large isotopic variations in this study exceed those that can be caused by errors in instrument accuracy and cannot be attributed solely to evaporation and condensation processes. Hydrogen and oxygen isotope anomalies can be used to trace hydrogeological processes that are closely related to seismically induced stress changes, such as changes in permeability, groundwater mixing, and water-rock interactions [51]. Therefore, we should give more consideration to the recharge sources of different fault systems. The $\delta^2$H variation in hot spring waters record mixing and conversion between different groundwater sources, while the δ18O shift usually occurs due to water-rock interactions in geothermal systems, resulting in the enrichment of heavier oxygen isotopes [52]. Earthquakes usually lead to large changes in hydrogen and oxygen isotopes; for example, $\delta^2$H has been shown to decrease by $-8$‰ and $\delta^{18}$O by $-1$‰ after the $Ms$5.8 earthquakes in Iceland [24]. Since inadequate and continuous data are available from before the earthquake, detailed dynamic processes involved in isotopic changes before the 2021 Yangbi $Ms$6.4 earthquake cannot be evaluated in detail. We argue that the strong variations in the stable isotopes $\delta^{18}$O and $\delta^2$H of hot springs were affected by water-rock interaction and mixing between different groundwater sources caused by switching between fracture pathways because of permeability change; it remains unclear when the abnormal isotope change appears before an earthquake.

**Table 1.** Results of stable oxygen and hydrogen isotope composition of hot springs and recharge elevation of hot spring.

| Hot Spring | Date | δ18O (‰) | δ2H (‰) | Recharge Elevation (km) | Variation Range | |
|---|---|---|---|---|---|---|
| | | | | | δ$^2$H (‰) | δ$^{18}$O (‰) |
| MD-01 | 2018/4/22 | −14 | −109.6 | 2.8 | −0.4 | −5.1 |
| MD-02 | 2021/5/25 | −14.4 | −114.7 | 2.9 | | |
| NJ-01 | 2017/2/28 | −13.4 | −95 | 2.3 | −0.2 | −19.1 |
| NJ-02 | 2021/5/26 | −13.6 | −114.1 | 2.9 | | |
| EYXX-01 | 2019/5/16 | −14.1 | −110.4 | 2.8 | −1.1 | −6.8 |
| EYXX-02 | 2021/5/26 | −15.2 | −117.2 | 3.0 | | |
| HYS-01 | 2017/2/28 | −13.5 | −96.2 | 2.3 | 0 | −22.4 |
| HYS-02 | 2021/5/26 | −13.5 | −118.6 | 3.1 | | |
| XSK-01 | 2017/2/28 | −13.3 | −100.6 | 2.5 | −0.8 | −17.4 |
| XSK-02 | 2021/5/26 | −14.1 | −118 | 3.0 | | |
| SB-01 | 2017/2/25 | −12.1 | −82.8 | 1.9 | −1 | −22.2 |
| SB-02 | 2021/5/25 | −13.1 | −105 | 2.6 | | |
| QG-01 | 2017/2/26 | −12.3 | −90.9 | 2.1 | −1.7 | −18.5 |
| QG-02 | 2021/5/25 | −14 | −109.4 | 2.7 | | |
| JXQ-01 | 2017/2.26 | −11.3 | −85.7 | 2.0 | −2.2 | −17.3 |
| JXQ-02 | 2021/5/24 | −13.5 | −103 | 2.5 | | |
| WBS-01 | 2017/2/25 | −12 | −89.5 | 2.1 | −0.3 | −21.61 |
| WBS-02 | 2021/5/24 | −12.3 | −111.11 | 2.8 | | |
| XCQ | 2021/5/25 | −13.6 | −105.5 | 2.6 | | |

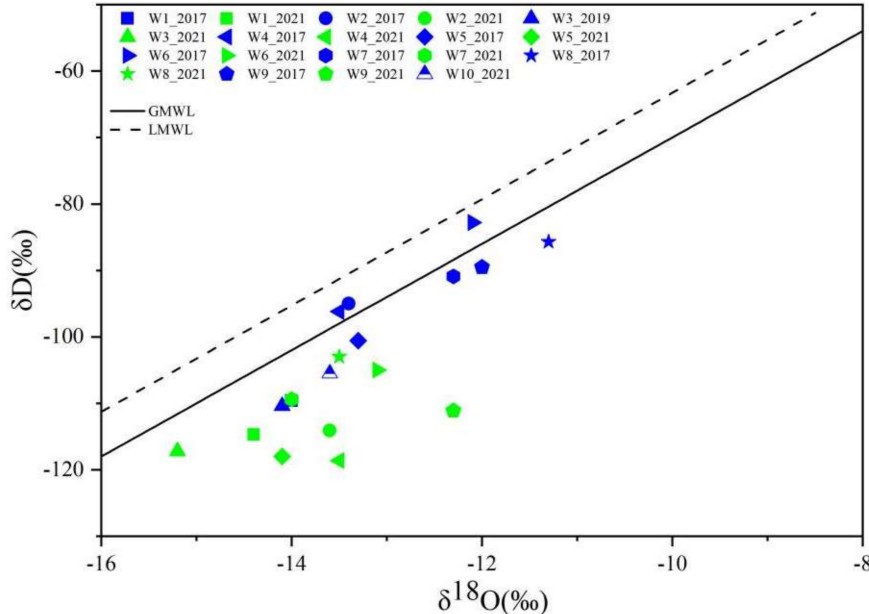

**Figure 2.** Stable oxygen and hydrogen isotope of the 10 hot springs and their correlations with GMWL and LMWL (the blue and green marks represent the observation results of the first and second phases, respectively) LMWL: δ$^2$H = 8.41δ$^{18}$O + 16.72.

### 4.2. Hydrochemical Characteristics of Hot Springs

#### 4.2.1. Dissolved Major Elements

Concentrations of Na$^+$, K$^+$, Ca$^{2+}$, and Mg$^{2+}$ ranged from 89.8250 to 1033.6580 mg/L, from 3.3940 to 47.4540 mg/L, from 3.7780 to 421.7830 mg/L, and from 0.8340 to 41.5160 mg/L, respectively. Concentrations of F$^-$, Cl$^-$, SO$_4^{2-}$, and HCO$_3^-$ ranged from 2.6280 to 9.4040 mg/L, from 3.7620 to 825.9770 mg/L, from 39.004 to 1580.1430 mg/L, and from 34.9550 to 2278.8260 mg/L, respectively. The hydrochemical types of most geothermal water are HCO$_3$SO$_4$-Na (W2, W3, W4, W5, W6, W7, W8, and W9), which are marked in the

C zone with green ellipses, except for two samples, those being $SO_4Cl$-NaCa (W1) in the B zone with blue ellipses and $SO_4$-Ca (W10) in the A zone with red ellipses (Figure 3). The hot spring samples have different total ionic salinity, TIS, ranging from 4.75 to 57.17 meq/kg, as indicated by the correlation plot of $Na^+ + K^+$ vs. $Ca^{2+} + Mg^{2+}$ [53] (Figure 4). MD (W1) spring's main chemical components were $Ca^{2+}$, $Na^+$, $SO_4^{2-}$, $Cl^-$, and $HCO_3^-$. The proportion of $Na^+$ and $Ca^{2+}$ was higher than 59% and 33%, whereas that of $SO_4^{2-}$ and $Cl^-$ was higher than 50% and 33%, respectively. Moreover, $HCO_3^-$ concentration is also relatively high, indicating that carbonatite and silicate rocks are involved in the water–rock reaction, as expressed in Equations (2), (3) and (5). However, the $Cl^-$ concentration of water samples from the W1 spring reached approximately 764.47–825.98 mg/L, suggesting the upwelling of deep-earth fluids into the spring. Additionally, this spring has the highest value of ionic salinity, TIS, with a value of 57.17. The remaining hot springs are of $HCO_3SO_4$-Na type and have much lower TIS (<20 meq/kg): their main chemical components include $Ca^{2+}$, $Mg^{2+}$, $Na^+$, $HCO_3^-$, and $SO_4^{2-}$. The latter usually comes from two sources: one is the sulfate rock containing gypsum minerals in the water–rock reaction environment, whereas the other is formed by products of the oxidation zone of sulfide deposits. Clastic sedimentary sandy conglomerates of the study area contain gypsum, and $SO_4^{2-}$ in hot spring water samples is related to the gypsum salt layer [54]. The sodium element ranks sixth in the content of the Earth crust, and it is widely distributed. Furthermore, granite, schist, sandstone, and limestone are well developed in the study area. These springs may have to percolate into deeper carboniferous carbonates, and thus the interaction of groundwater with granite and carbonate may have formed the chemical characteristics of the hot spring water, as shown in Equations (2)–(5). More in detail, the main chemical components of the XQC spring were $Ca^{2+}$, $Mg^{2+}$, $SO_4^{2-}$, and $HCO_3^-$. The proportion of $Ca^{2+}$, $SO_4^2$, and $HCO_3^-$ was higher than 74%, 77%, and 22%, respectively. This spring has the higher value of ionic salinity, TIS, with a value of 37.20. The consistent enrichment of sulfate and calcium as the main feature of $SO_4$-Ca water formation may be mainly due to the dissolution of gypsum and anhydrite during infiltration. In general, Ca/Mg-rich minerals, such as calcite, dolomite and mica, play an important role in hot spring water–rock interactions. The W10 hot spring is enriched in $Ca^{2+}$ with excessive $SO_4^{2-}$, and $HCO_3^-$, implying the involved dissolution of calcite and dolomite, as expressed in Equations (2)–(4). Furthermore, it has been found that the hydrochemical types of hot springs did not significantly change.

$$2NaAlSi_2O_3 + 3H_2O + CO_2 \rightarrow H_2Al_2Si_2O_3 \cdot H_2O + 4SiO_2 + 2Na^+ + HCO_3^- + OH^- \tag{2}$$

$$CaCO_3 + H_2O + CO_2 \rightarrow Ca^{2+} + 2HCO_3^- \tag{3}$$

$$MgCO_3 + H_2O + CO_2 \rightarrow Mg^{2+} + 2HCO_3^- \tag{4}$$

$$CaSO_4 \cdot H_2O \rightarrow Ca^{2+} + SO_4^{2-} + H_2O \tag{5}$$

### 4.2.2. The Water–Rock Reaction Equilibrium

The water-rock equilibrium state of hot water is often illustrated by the Sodium-Potassium-Magnesium triangle, established by using relative Na/1000, K/100, and $Mg^{1/2}$ contents, which is roughly divided into three stages: complete equilibrium water with a high reaction degree, partial equilibrium water with a medium reaction degree and immature water in the process of rock dissolution and leaching [55]. The Na-K-Mg triangular plot shows that W4 plot in the partial equilibration zone, W1 in the immature water zone, whereas other hot spring sample plots in the intersection line between the two (Figure 5). This phenomenon demonstrates that the reaction between groundwater and the surrounding rock is insufficient; thus, hot water will be cooled to a certain extent in the process of rising to the surface. It may also occur that, when flowing through the rock, the proportion of immature surface cold water is relatively large, so that both water temperature and salinity of hot spring water become low, resulting in the difficulty of water sample reaching equilibrium.

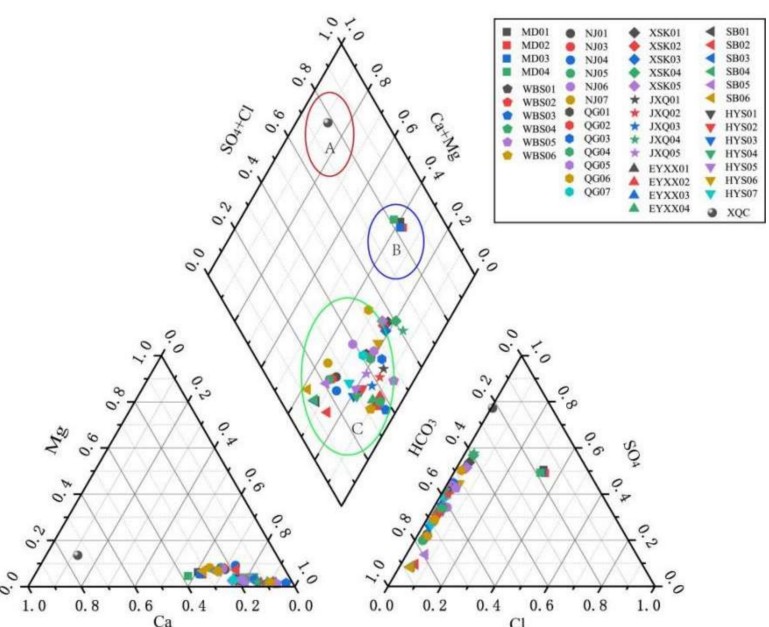

**Figure 3.** Piper diagram of the spring water samples along WQF. The hot spring samples can be divided into 3 clusters according to the main ions, which are plotted in A (red ellipses), B (blue ellipses), and C (green ellipses) blocks.

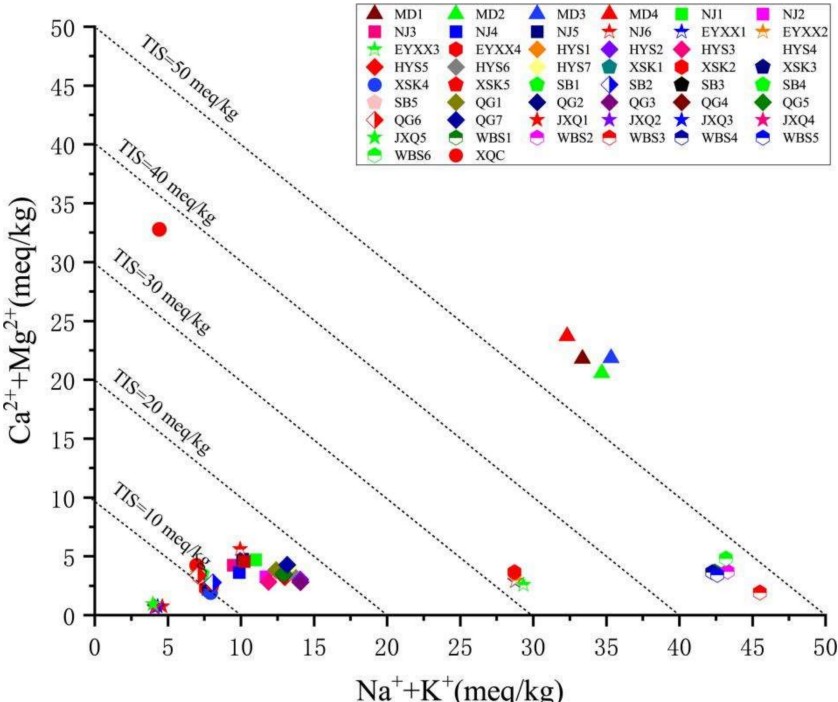

**Figure 4.** Correlation plot of Na$^+$ +K$^+$ vs. Ca$^{2+}$ +Mg$^{2+}$ for the spring water samples along WQF, also showing iso-ionic-salinity (TIS) lines for reference.

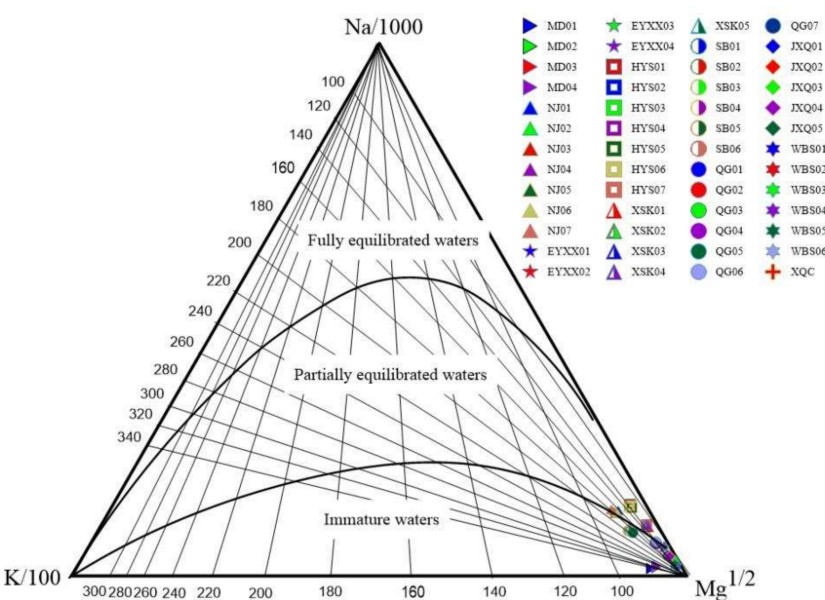

**Figure 5.** Distribution of aqueous samples on the Na/1000-K/100-Mg$^{1/2}$ ternary diagram.

### 4.2.3. Mineral Saturation States

Regarding the analysis of hot spring water samples, the mineral saturation index (SI) is calculated by using PHREEQC software. The results of this study are presented in Figure 6: it can be noticed that all spring water samples are saturated (SI > 0) with respect to goethite, aragonite, dolomite, quartz, barite, and calcite, while chalcedony and fluorite are basically in equilibrium (SI ≈ 0). However, SI with respect to talc varies greatly in each hot spring water. Groundwater samples in W2, W5, W6, W7, and W8 are in a supersaturated state (SI values are 9.07, 7.98, 4.03, 2.4, and 4.15, respectively) and almost in equilibrium if considering W4 (SI = 0.48). However, they are unsaturated in W01, W03, W09, and W10 (SI values are −2.48, −1.72, −0.96, and −1.82, respectively). This phenomenon may reflect the difference in the surrounding rock characteristics. The supersaturation indicates high content of these minerals and long residence time in the aquifer system [56]. However, albite, alunite, anglesite, anhydrite, anorthite, chrysotile, gypsum halite, manganite, and pyrochroite are found in an unsaturated state in the majority of spring waters, indicating that they are relatively soluble or have insufficient reaction time with hot water.

### 4.2.4. Hot Water Thermal Storage Temperature and Circulation Depth

Thermal storage temperature is an important parameter for evaluating geothermal resources [57–60]. Silica geothermal temperature scale is the earliest and most commonly used geochemical temperature scale, based on the theory that the silica content in geothermal fluids depends mainly on the solubility of quartz in water at different temperatures and pressures. In this study, the quartz silica temperature scale is selected, as in Equation (6).

$$T = \frac{1309}{5.19 - \log(SiO_2)} - 273.15 \tag{6}$$

where $SiO_2$ is the $SiO_2$ concentration in the hot spring samples, and the results are shown in Table 2. The reservoir temperatures of the hot spring samples in the study area were calculated to be 44.1 °C–101.1 °C. The circulation depth was estimated based on the local geothermal gradient and the thermal storage temperature of the hot spring water, as in Equation (7).

$$H = (T - T0)/q + h0 \tag{7}$$

where $H$ is the circulation depth (km), $h0$ is the depth of the thermostatic zone (km), $T$ is the reservoir temperature (°C), $T0$ is the temperature of constant temperature zone

($^\circ$C), and $q$ is the geothermal gradient ($^\circ$C/km) [61]. Considering previous studies on groundwater in some areas of Yunnan Province, a geothermal gradient q of 20 $^\circ$C/km, an annual average temperature $T0$ of 15.8 $^\circ$C, and constant temperature zone depth $h0$ of 30 m were assumed [62]. Therefore, the final circulation depth of the 10 hot springs ranged from 1.4 to 4.3 km, while the average circulation depth was 3.2 km (Table 2). In addition, we can see that the W1 circulation depth in the northern section and the W10 circulation depth in the southern section are shallow, with values of 1.4 and 1.7, respectively, compared to the other hot springs in the middle section.

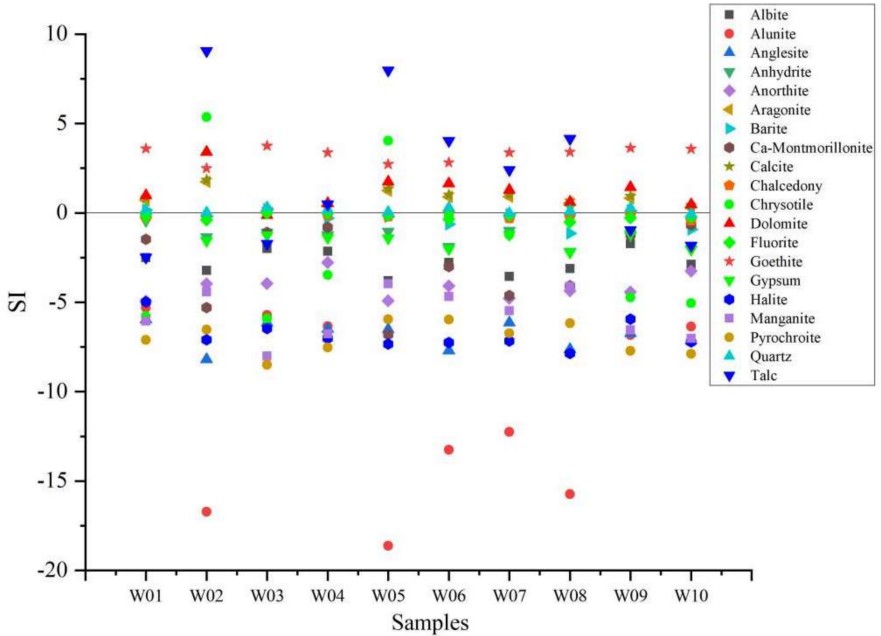

**Figure 6.** Saturation indices values of groundwater samples with respect to minerals.

**Table 2.** Analytical data of Reservoir Temperature and Circulation Depth.

| No. | Site | SiO$_2$ (mg/L) | Reservoir Temperature ($^\circ$C) | Circulation Depth (km) |
|-----|------|----------------|-----------------------------------|------------------------|
| W1 | MD | 11.6 | 44.1 | 1.4 |
| W2 | NJ | 33.1 | 83.5 | 3.4 |
| W3 | HYS | 48.8 | 100.7 | 4.3 |
| W4 | EYXX | 37.9 | 89.3 | 3.7 |
| W5 | XSK | 49.2 | 101.1 | 4.3 |
| W6 | SB | 46.4 | 98.4 | 4.2 |
| W7 | QG | 27.5 | 75.9 | 3.0 |
| W8 | JXQ | 26.3 | 74.1 | 2.9 |
| W9 | WBS | 22.4 | 67.8 | 2.6 |
| W10 | XCQ | 13.7 | 49.8 | 1.7 |

### 4.3. Dissolved Trace Element

4.3.1. Strontium Isotope

The tritium isotopic composition in hot spring waters can reflect the lithologic characteristics of the strata where they flow. The Sr (strontium) concentration, and the values of $^{87}$Sr/$^{86}$Sr varied from 0.254 to 12.534 mg/L, and from 0.7080 to 0.7184, respectively. The Sr concentration in W1 and W10 springs was approximately 9.6620 and 12.5340 mg/L, respectively. The $^{87}$Sr/$^{86}$Sr ratio of carbonate and sulfate weathering sources is approximately 0.7080, while the one of aluminum silicate is approximately ranging from 0.7160 and 0.7200 (Figure 7). Therefore, except that the W8 hot spring belongs to the silicate mineral weathering, the remnant springs are between carbonate and silicate mineral weathering,

indicating that they are formed by the interaction with Sr-bearing rocks in the crust during the deep circulation of atmospheric precipitation in the local heat flow system, consistently with the results of hot spring chemistry.

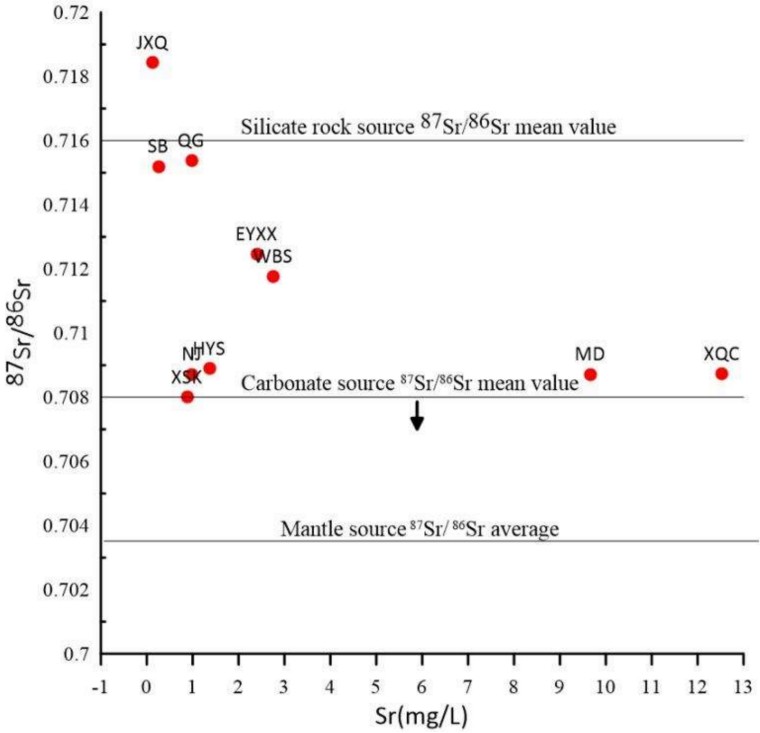

**Figure 7.** Strontium isotopic composition of 10 hot springs.

### 4.3.2. Trace Element Origin

The content of trace elements in geothermal water and the formation process are related to mineral composition, rock properties and geothermal activity, and to some extent can reflect the degree of water–rock interaction. The enrichment factor is one of the important indicators for quantitative evaluation of trace element source and it selects elements that meet certain conditions as reference elements (or standardized elements), which is expressed by Equation (8):

$$EF\text{i} = \frac{(Ci/Ti)w}{(Ci/Ti)r} \tag{8}$$

where $C_i$ is the content of an element in the sample, $Ti$ is the content of the selected reference element (Ti), $w$ is the concentration of the element in the water sample, and $r$ is the concentration of the element in the rocks of the study area. In this study, 19 types of trace elements were measured, including Li, Be, Ba, Sr, Ti, V, Cr, Mn, Fe, Co, Ni, Cu, Zn, Mo, Tl, Pb, U, Sn, and Al. Spring waters of the WQF are compared with the corresponding trace element average compositions of the multiple phase granitic rocks in the northern and middle segment of DianCangShan–AiLaoShan. The rock chemistry is discussed in the reference of Ji et al. [63]. The $EF_i$ of Mn is higher in respect to others (Figure 8); this element is widely distributed on Earth and is more common in granite. Generally speaking, hot spring water mainly contains the carbonatite of divalent Fe and Mn, which are dissolved by hot spring waters. In addition, except for W1 and W8, Li is enriched: its concentration in W9 springs is up to 2047 µg/L, while in W2, W3, W5, and W6 hot springs, it is over 1000 µg/L. Li is an active element with high hydrolysis energy, which is easily enriched in hot spring water and is the signature element of deep liquid upwelling [64]. Volcanic rocks are widely distributed in the study area, so the Li content in hot springs around the

fault zone is high. In addition, due to the activity characteristics of Li, it is likely to become a symbolic element of deep liquid upwelling during deep fault activity. In addition, Sr is relatively enriched in W1, W6, W9, and W10. Sr is an alkaline earth metal dispersed element, abundant in the crust and mantle, and its migration is closely related to Ca. In addition, Sr is more easily enriched in weak alkaline water with pH 7.0–8.5 [65]. Due to the carbonate and clastic components developed around the Weixi–Weishan area, the pH value of spring samples ranged from 6.62 to 8.24, which is typical of weak-alkaline waters. The overall $EF_i$ and the content of trace elements are relatively low: the Na-K-Mg triangle showed that the hot spring waters in the area are in the intersection line of partial equilibration zone and immature water zone, thus indicating that the degree of water–rock reaction was relatively weak in the study area.

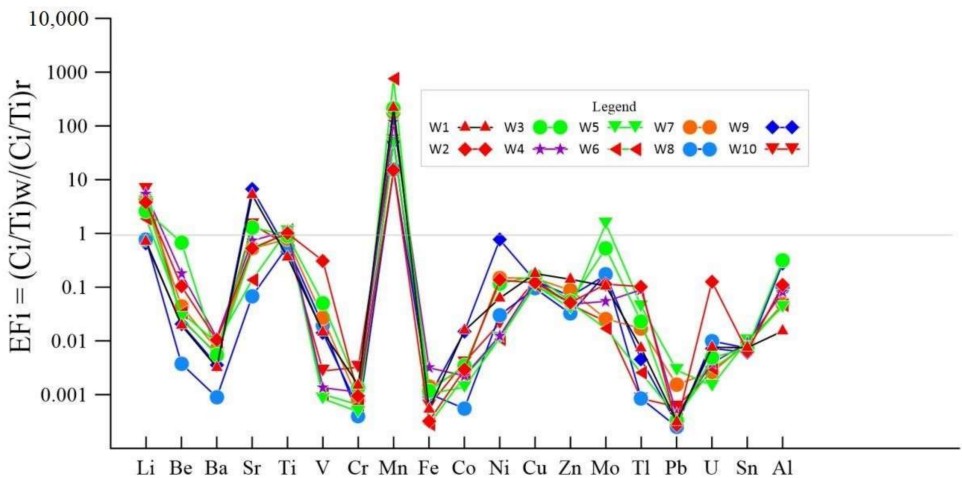

**Figure 8.** Trace element distribution map of WQF hot springs in terms of enrichment factor (weight ratio) in Ti.

### 4.4. Coupling between Chemical Variations in Spring Waters and Earthquakes

There have been three earthquakes with $Ms \geq 5.0$ in the study area since 2015: the Eryuan $Ms5.0$ earthquake on 18 May 2016, the Yangbi $Ms5.1$ earthquake on 27 March 2017, and the Yangbi $Ms6.4$ earthquake on 21 May 2021 (Figure 1b). In annual continuous data of spring waters, abnormal variations were detected in springs W2(NJ), W3(EYXX), W6(SB), W7(QG), and W9(WBS) in 2017, of which W2(MD) and W3(NJ)are located to the north of the earthquake, while the others are to the south, which could be considered as the result of the influence of the Yangbi $Ms5.1$ earthquake, as they were detected in the absence of extreme weather, such as precipitation changes, which were detected when precipitation did not occur. The anomaly characteristics and epicentral distance in the hot springs are shown in Table 3. Springs W2 and W3, which are 44–45 km away from the epicenter, showed that $Na^+$, $Cl^-$, and $SO_4^{2-}$ significantly decreased before the earthquake: $Na^+$ in W2(NJ) decreased by 23.80%, while $Na^+$, $Cl^-$, and $SO_4^{2-}$ in W3 decreased by 15.10%, 13.70%, and 11.20%, respectively, from April 2016 to February 2017. However, it is worth noting that to the south of the earthquake, $Na^+$, $SO_4^{2-}$, and $Cl^-$ of W6(SB), W7(QG), and W9(WBS) showed a significant increase before the earthquake. Moreover, $Na^+$, $Cl^-$, and $SO_4^{2-}$ in W6, which is 16 km away from the epicenter, increased by 10.82%, 28.45%, and 14.45%, respectively, while in W9, 100 km away, they increased by 5.0%, 12.33%, and 14.84%, respectively; $Na^+$ and $SO_4^{2-}$ in W7, 54 km away, increased by 8.95% and 6.62% from April 2016 to February 2017, respectively (Figure 9).

**Table 3.** The occurrence time of precursory anomalies in the continuous monitoring sites before three earthquakes.

| Name | Time Period | Rate of Change | | | Distance | Structural Location to Earthquake |
|------|-------------|----------------|------|------|----------|-----------------------------------|
| | | Na$^+$ | Cl$^-$ | SO$_4{}^{2-}$ | | |
| NJ | 2015/3/19–2016/4/2 | 13.59% | −13.29 | NAN | 44 | Northern Segment |
| | 2016/4/20–2017/2/29 | −23.80% | 11.12 | NAN | | |
| | 2017/2/29–2017/3/29 | 4.02% | −2.53% | −2.60% | | |
| | 2017/3/29–2018/3/27 | 2.31% | 0.35 | 4.30% | | |
| | 2018/3/27–2019/5/16 | 1.31% | 2.97% | 2.04% | | |
| | 2019/5/16–2021/5/26 | 3.14% | 20.45 | 1.76 | | |
| HYS | 2015/3/19–2016/4/18 | 0.39% | −6.83% | 0.39% | 43 | Southern Segment |
| | 2016/4/18–2017/2/28 | −15.10% | −13.70% | −11.20% | | |
| | 2017/2/28–2017/3/29 | 9.06% | 9.62% | 7.20% | | |
| | 2017/3/29–2018/3/27 | 0.47% | 1.38% | 4.17% | | |
| | 2018/3/27–2019/5/16 | 2.85% | 2.56% | 4.72% | | |
| | 2019/5/16–2021/5/25 | −0.67% | 5.67% | 6.75% | | |
| SB | 2016/4/18–2017/2/26 | 10.82% | 28.45% | 14.45% | 16 | Southern Segment |
| | 2017/2/26–2017/3/28 | −11.55 | −26.89% | −13.69% | | |
| | 2017/3/28–2018/3/26 | 1.16% | −1.22% | 6.40% | | |
| | 2018/3/26–2019/5/17 | 0.12% | 3.17% | 1.37% | | |
| | 2019/5/17–2021/5/26 | −1.81% | 15.25% | 8.60% | | |
| QG | 2015/3/20–2016/4/17 | 4.29% | 25.35% | 6.62% | 54 | Southern Segment |
| | 2016/4/17–2017/2/26 | 8.95% | −3.72% | 6.62% | | |
| | 2017/2/26–2017/3/28 | −8.14% | −14.47% | −16.40% | | |
| | 2018/3/26–2019/5/17 | 2.28% | 14.54% | 2.78% | | |
| | 2019/5/17–2021/5/24 | 0.90% | 1.55% | 4.00% | | |
| WBS | 2015/3/16–2016/4/3 | 2.07% | 1.46% | 3.43% | 100 | Southern Segment |
| | 2016/4/3–2017/2/25 | 5.00% | 12.33% | 14.84% | | |
| | 2017/2/25–2018/4/20 | −7.04% | −12.30% | −12.85% | | |
| | 2018/4/20–2019/5/17 | 0.87% | −1.35 | 2.58% | | |
| | 2019/5/17–2021/5/24 | 1.12% | 10.91% | 1.76% | | |

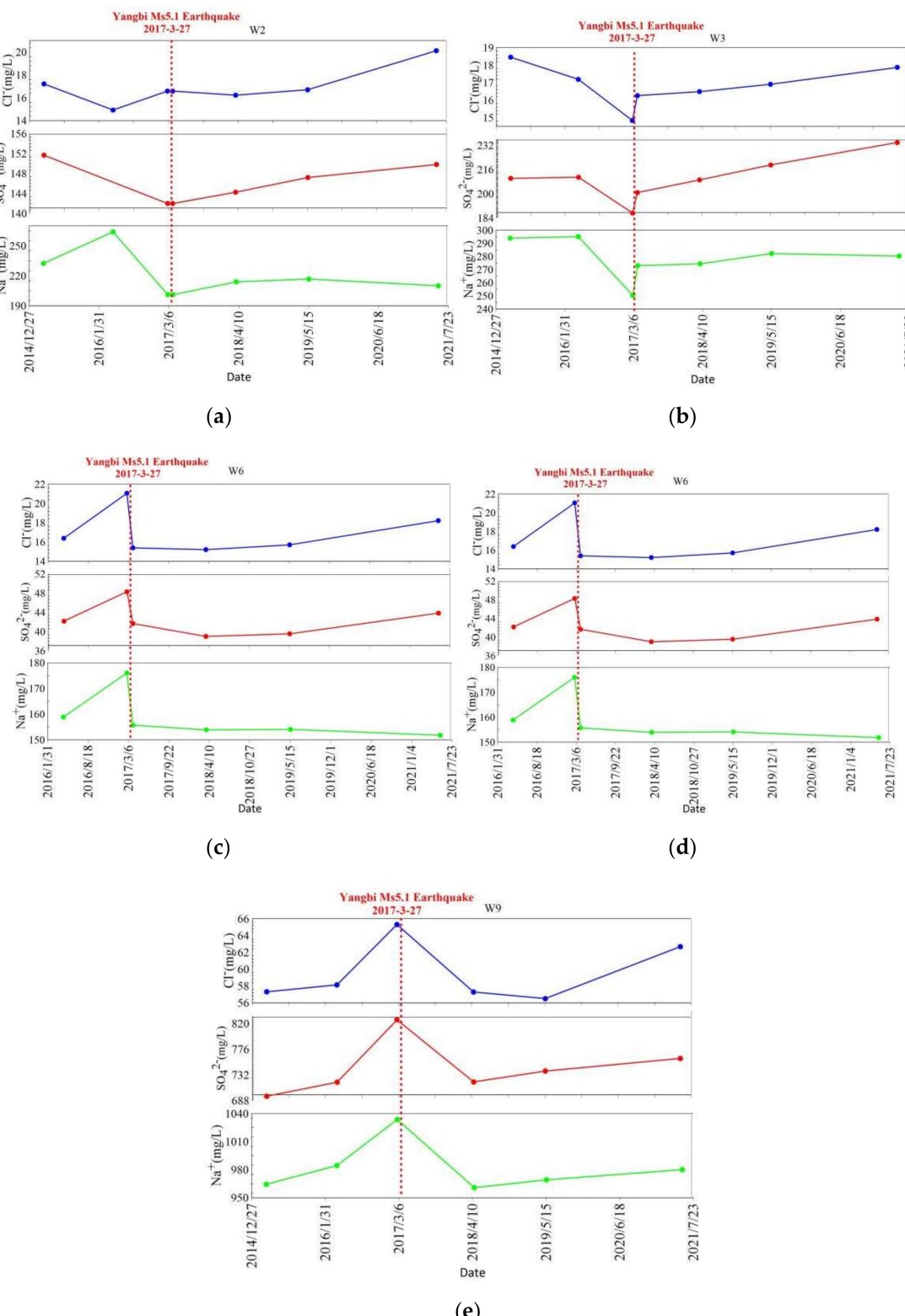

**Figure 9.** The variation characteristics of concentration of Na$^+$, Cl$^-$, SO$_4^{2-}$ and related to Yangbi *Ms*5.1 earthquake in 27 March 2017. (**a**,**b**) are in the north of the earthquake and (**c**–**e**) are in the south of the earthquake.

Typically, the mechanism by which groundwater is altered by earthquakes is due to changes in the stress-strain state of the fracture zone caused by the earthquake, resulting in changes in aquifer properties and local hydrostatic release due to rock spreading, as

well as mixing of water from different aquifers and release of deep geothermal fluids [49]. The effects of earthquakes on strain or stress are usually static and dynamic, with the former manifesting itself as the effect on fault activity during seismogenic processes and the latter as the effect of seismic waves on aquifers. That means the main requirements for the occurrence of an earthquake are significant strain energy accumulation and a sudden stress drop. When strain accumulation occurs along fault sections and the original physicochemical equilibrium of the Earth crust is altered, this results in changes in the water–rock reaction degree and in microfractures in the basement rock in hydrochemistry, and other related parameters [6,7], resulting in mixtures of different waters and in changes in concentration of hydrochemistry components until the earthquake occurred.

The five analyzed hot springs are located in the WQF and sensitive to seismicity. However, there are differences in the major ion changes between the north and south before the Yangbi *Ms*5.1 earthquake, mainly reflected in the fact that the hot spring ions were mainly decreasing in the north before the earthquake, while they were increasing in the south (Figure 10). This may be related to the seismogenic mechanism of earthquakes and the difference in seismic activity of fault zones. Although the strong seismic activity of the WQF is not significant in history, the largest earthquake in the area is the Shanglan M $6^1/4$ earthquake that occurred in the Maden basin in 1948 in the northern segment. According to the latest research results, ancient earthquake relics have been revealed in the vicinity of the Tongdian basin and Yushichang, located in the northern segment as well. In addition, the Eryuan *Ms*5.5 and *Ms*5.0 earthquakes in 2016 occurred in the northern section of the fault zone [66]. Then, the Yangbi *Ms*5.1 earthquake occurred on 27 March 2017. Moreover, this study also shows that the hot spring water circulation depth in this section is deeper than in south segments. In addition, recent studies have shown that on the northern segment, the dextral horizontal slip rate is 1.8–2.4 mm/year, while the vertical slip rate is 0.3–0.35 mm/year, and the average horizontal sliding rate is 1.25 mm/year on the east side of the city of Yangbi County [46]. Therefore, according to these analyses, the present seismic activity characteristics of the WQF zone are the highest in the northern segment, as compared with the southern. Based on the sturdy body seismogenic model, we assumed that the northern segment is in a relatively locked state under the corresponding regional stress [67–73]. The difference in fault activity characteristics will alter the volume and permeability of the medium. The deeper locked and unstable state of the fault may accelerate the fluid movement from the deep rock medium to the aquifer along the fault and then into the surrounding area, which may result in water mixing, hydrogeochemical changes, and the deviation of the original equilibrium state of the system, thus causing the content of hydrogeochemical ions to decrease [74]. Meanwhile, it has been observed that the closer the epicentral distance is, the larger are the anomalies that appear. The $Cl^-$ concentration is more sensitive to near-field seismicity than $Na^+$ and $SO_4^{2-}$ (W6). A similar observation was reported by Li et al. for the Xiaojiang fault: $Cl^-$ concentration seems in good agreement with near-field seismic activity within 50 km and with a magnitude range from $M_L$1.0 to 4.0 [75]. In 1995, $Cl^-$ in the water collected in a well located 20 km away from the Kobe earthquake in Japan also increased significantly [76]. Five days before the 1996 $M_L$5.2 earthquake in the French Pyrenees, $Cl^-$ concentrations in nearby springs increased by 36.0% over background values [77]. When the earthquakes occurred, the pore pressure in the rocks would have been enhanced, whereas the seismic wave propagated through. $Cl^-$ is mainly derived from the deep earth and the source is relatively single, which is less affected by mixing shallow cold water, suggesting the contribution of upwelling deep-earth fluids into the spring.

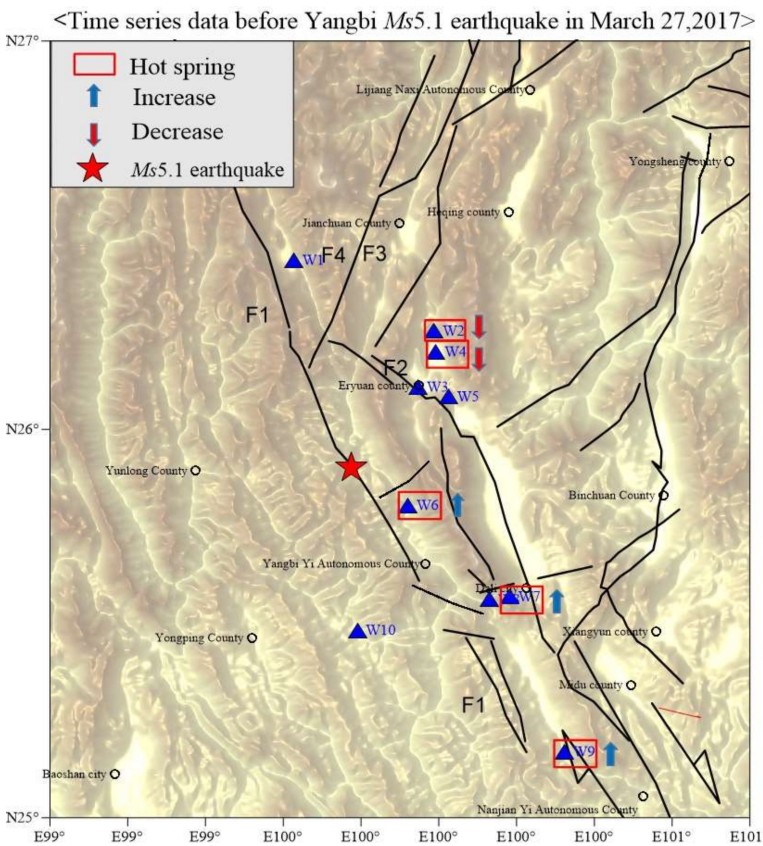

**Figure 10.** Summary of anomalies in hot spring before Yangbi *Ms*5.1 earthquake on 27 March 2017 based on overall results.

In addition, continuous measurements were collected every three days in two hot springs, including NJ (H1) since December 2018 and EYXX (H2) since March 2021. During the observation period, there were no earthquakes with an Ms ≥ 4.0 except the Yangbi *Ms*6.4, but there were several earthquakes with *Ms* ≥ 3.0 instead. It was shown that there were some irregular fluctuations of the major elements ($Na^+$, $Cl^-$ and $SO_4^{2-}$) before and after the earthquake. Here, we focus on the relationship between the Yangbi *Ms*6.4 earthquake and the change in hot spring waters. In H01, a change in $Na^+$, $SO_4^{2-}$, and $Cl^-$ showed obvious decreasing trends in early March, before two Eryuan *Ms*3.8 earthquakes occurred on 1 March and 18 April 2021. In late April and early May, approximately 20 days before the Yangbi *Ms*6.4 earthquake, $SO_4^{2-}$, and $Cl^-$ showed an obvious upward trend (Figure 11a). In H2, in the same time period as H1 in late April and early May, $Na^+$, $SO_4^{2-}$, and $Cl^-$ showed a turn up after decline (Figure 11b), although the change is not very large. It is worth noting that the irregular variations observed before the Yangbi *Ms*6.4 earthquake showed synchronous anomalies, supporting a possible relationship with the target earthquake; we link this irregular variation with the mixing between groundwater components, which Skelton et al. attributed to preseismic dilation [23,24]. The two springs are located 45–50 km away from the epicenter of the earthquake but not in the seismogenic fault zone. Li et al. (2021) did not find obvious surface deformation along the WQF, related to the Yangbi *Ms*6.4 earthquake, through field investigation in the seismic area, combined with the comprehensive analysis of focal mechanism solution, aftershock distribution, and In-SAR inversion results. Thus, it has been judged that the Weixi–Qiaohou main fault has not been related to this *Ms*6.4 earthquake. Considering the tectonic location and distance from the epicenter, it can be inferred that the response of hot springs to seismic events is not only related to the magnitude and distance from the epicenter, but is also controlled by the tectonic stress of deep major faults. The irregular variations in H1 and

H2, relative to the Yangbi *Ms*6.4 earthquake, confirmed the anisotropic characteristics of stress/strain transmission through the structural discontinuity between the focal area and the measurement point. This result also shows the sensitivity of hydrogeochemical characteristics to the seismic response.

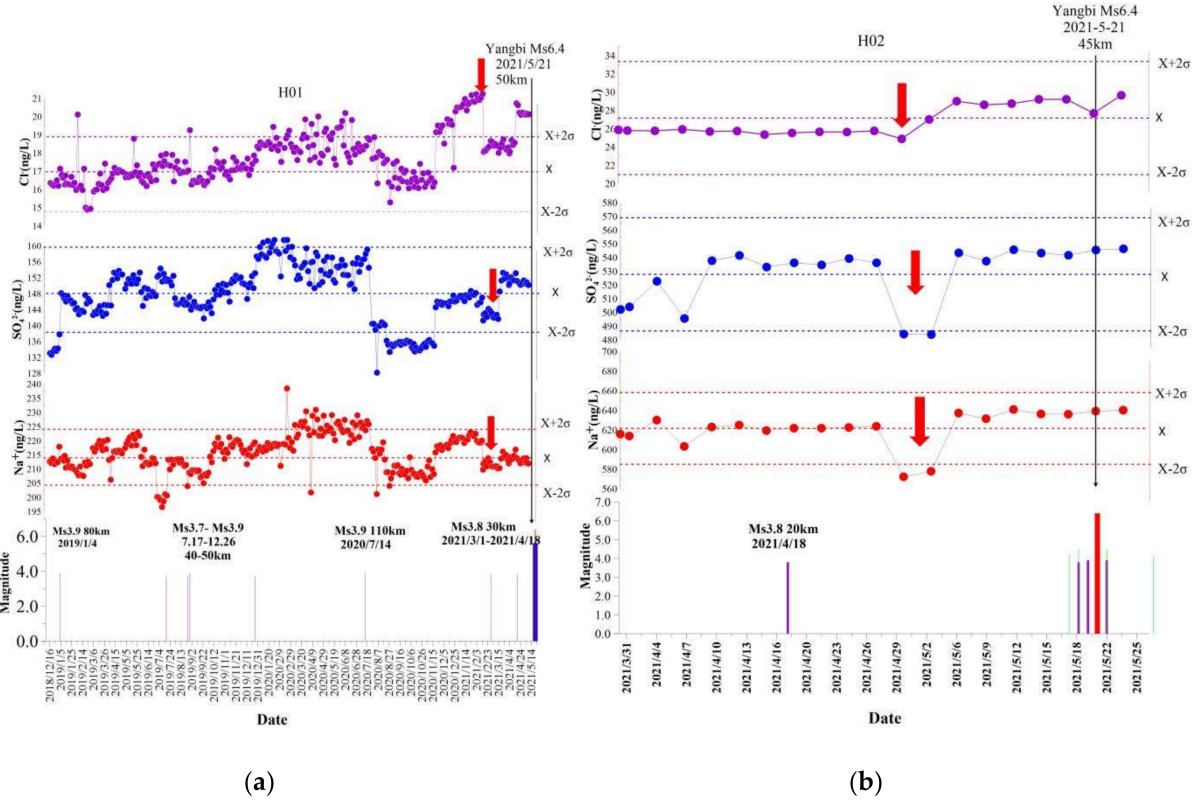

(**a**)　　　　　　　　　　　　　　　　　　　(**b**)

**Figure 11.** Temporal variations in concentration of $Na^+$, $Cl^-$, $SO_4^{2-}$ and related earthquake. (**a**) is stand for NJ spring; (**b**) is stand for EYXX spring. Purple bars show the earthquakes of $3.0 \leq Ms < 4.0$ within 120 km. Green bars show the earthquakes of $4.0 \leq Ms < 5.0$ within 150 km. Red bars show the earthquakes of $Ms \geq 60$ within 300 km. There were no earthquakes of $5.0 < Ms \leq 6.0$ during the observation period.

### 4.5. The Hydrogeochemical Circulation Model of Hot Spring Waters in the Weixi–Qiaohou Fault

Understanding the origin and migration pathways of spring waters in an active fault zone is extremely important in terms of studying hydrogeochemical earthquake precursors. According to our results, a conceptual model for the origin and the hydrogeochemical cycling process of hot spring waters in the WQF is presented in Figure 12. At an altitude of 1.9–3.1 km, the hot springs were recharged by infiltrated precipitation into aquifers along the fault between and around mountains and river terraces through the water-conducting fault zone. Studies have shown that there are distinct high electric conductivity and low-velocity materials from the Songpan–Ganzi block to the southern Sichuan–Yunnan diamond block [78]. In addition, in the area with a high helium isotope ratio of hot spring gas in the active fault zone, there is usually an obvious high conductivity and low velocity area. $^3He/^4He$ values of hot spring gas samples are less than 0.7 Ra in the Jinshajiang–Red River fault including the WQF, which indicates that such a fault area is the main channel for helium degassing in the crust [79]. Areas of high conductivity and high velocity in the fault zone are considered to be zones of fault disruption caused by impact sliding or shearing along the fault and filled with deep source magma and/or metamorphic fluids. Then, the water was heated to 44.1 °C–101.1 °C when the circulation depth of groundwater reached 1.4–4.3 km. Under certain temperature and pressure conditions, water–rock reactions occur at different depths. Since the degree of water–rock reaction is not particularly strong, all hot

spring waters are partially in equilibrium and immature. During the cyclic rise, they may mix with colder surface water or shallow groundwater and eventually become hot springs that appear at the surface. Strong tectonic movements and seismic activity will break the equilibrium state between water and rock in the crustal medium, and this can cause changes in the local stress field of the fault zone, resulting in changes in the equilibrated state of hot spring waters and the hydrochemical information carried by waters, which can provide evidence for short-term and imminent earthquake prediction.

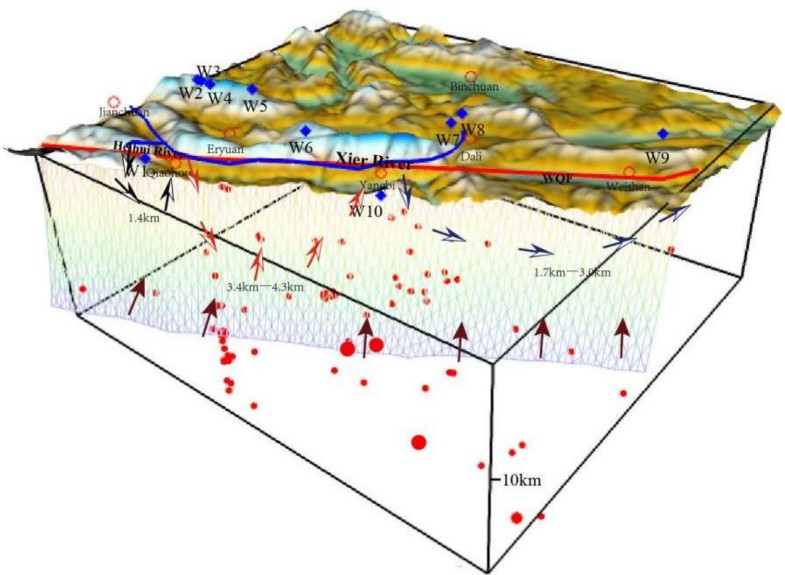

**Figure 12.** Conceptual model of the origin of groundwater and the hydrogeochemical cycling process in the WQF. Circulation depths were of 1.4 km in the northern section (black arrow) and 3.4 to 4.3 km in the middle section (black arrow), and 1.7 to 3 km in the southern section (blue arrow). The thick blue line represents the river and blue line represents WQF. Red-brown arrows refer to possible circulation in the deep part of the fracture zone.

## 5. Conclusions

We investigate the hydrogeochemical characteristics of 10 hot spring sites located along the WQF by investigating their major and trace elements as well and hydrogen and oxygen isotopes, as well as the characteristics of changes in the hydrogeochemical characteristics of hot springs relative to earthquakes in the region, especially the Yangbi *Ms*5.1 earthquake on 21 March 2017, and the Yangbi *Ms*6.4 earthquake on 21 May 2021. The following preliminary conclusions can be drawn from the study of the hydrogeochemical characteristics of hot springs in the region. The hot spring waters in the WQF were recharged by infiltrated precipitation, and the recharge elevation range is 1.9–3.1 km. The hydrochemical types of most geothermal waters, mainly controlled by aquifer lithology, are $HCO_3SO_4$–Na (W2, W3, W4, W5, W6, W7, W8, and W9), except for two samples being $SO_4Cl$–NaCa (W1) and $SO_4$–Ca (W10). The heat storage temperature range was inferred from an equation based on $SiO_2$ concentration and chemical geothermal modeling as 44.1 °C–101.1 °C. The depth of circulation of the thermal springs varies between 1.4 and 4.3 km. Compared with the middle section, the hot spring circulation depth of the northern and southern sections is shallower.

From the study of coupling between chemical variations in spring waters and earthquakes we can draw the following conclusions. First, based on hot springs with annual continuous data analysis, results presented a good correlation between hydrochemical compositions anomalies and the Yangbi *Ms*5.1 earthquake that occurred on 21 March 2017. In particular, $Na^+$ and $SO_4^{2-}$ decreased in the northern section before the earthquake, but the opposite trend has been observed in the southern section. This may be related to the seismogenic mechanism of earthquakes and the difference in seismic activity of fault zones.

Comparing the characteristics of all ions related to earthquakes, $Cl^-$ concentration is more sensitive to near-field seismicity than $Na^+$ and $SO_4{}^{2-}$. Then, the continuous measurement results of NJ(H1) and EYXX(H2) presented irregular variation anomalies 20 days before the Yangbi *Ms*6.4 earthquakes. Nevertheless, the abnormal range is not very large; thus, we inferred that the response of hydrogeochemical anomalies in hot springs to seismic events is not only related to the magnitude and epicenter distance but may also be controlled by the tectonic stress of deep and large faults and may also be related to the difference in activity of the fault zones.

According to the results, a conceptual model for the origin and the hydrogeochemical cycling process of hot spring waters in the WQF is shown. Based on the monitoring data and study results, the changes we observed associated with the WQF seismics may be caused by water–rock interactions, shallow and deep aquifer mixing, deep fluid upwelling, and bedrock fracture opening. Although these changes are specific to earthquakes examined in this study, we infer that chemical and isotopic signals of dilation might be detected elsewhere before earthquakes. We do not assume that this method can predict earthquakes; instead, we highlight groundwater chemistry as a promising target for future earthquake prediction studies.

**Supplementary Materials:** The following are available online at https://www.mdpi.com/article/10.3390/w14010132/s1, Table S1: Physicochemical parameters and analytictableal data of major elements of the spring waters, Table S2: Analytical data of trace elements on hot spring water sample.

**Author Contributions:** Conceptualization, H.Z. and X.Z.; methodology, H.Z., X.Z., H.S., Y.L.; software, H.Z., X.Z. and R.B.; validation, Y.Y., S.O. and F.L.; formal analysis, H.Z.; investigation, H.Z., X.Z., S.O. and F.L.; data curation, H.Z. and X.Z.; writing—original draft preparation, H.Z.; writing—review and editing, H.Z. and X.Z.; visualization, H.Z.; supervision, X.Z. All authors have read and agreed to the published version of the manuscript.

**Funding:** The work was funded by Field station fund of Gansu Seismological Bureau (2021Y16), Gansu Youth Science and Technology Fund (No.:**21JR7RA796;20JR10RA500**);Basic scientific research business project of Institute of earthquake prediction, China Seismological Bureau (No.**2021IESLZ05**); Key project of spark plan of China Seismological Bureau (No.:**XH21033**);National Key Research and Development Project (**2017YFC1500501**, **2019YFC1509203**) and the National Natural Science Foundation of China (**41673106**, **42073063**, **4193000170**) and The Special Fund of the Institute of Earthquake Forecasting, China Earthquake Administration (**2018IEF010104**, **2019CSES0104**, **2020IEF0604**, **2020IEF0703**, **2021IEF0602**, **2021IEF0101**, **2021IEF1201**), Spark Program for Earthquake Science and Technology (**XH20066**).

**Informed Consent Statement:** Informed consent was obtained from all subjects involved in the study.

**Data Availability Statement:** The raw data supporting the conclusions of this article will be made available by the authors, without undue reservation.

**Acknowledgments:** The authors are grateful to the Editor and anonymous reviewers for their constructive comments and suggestions.

**Conflicts of Interest:** The authors declare no conflict of interest.

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
