# Peer review of "Hydrochemical Characteristics of Earthquake-Related Thermal Springs along the Weixi–Qiaohou Fault, Southeast Tibet Plateau"

_water, doi:10.3390/w14010132_

Round 1

Reviewer 1 Report

Dear authors,

the study is very interesting with great utility in the application of different methodologies to describe the relationship between earthquakes and hydrogeology.

I have made different suggestions and recommendations in the pdf to improve the manuscript.

Best wishes.

Reviewer 2 Report

Water

Manuscript Number: 1507814

Title: Hydrochemical characteristics of earthquake-related thermal springs along the Weixi-Qiaohou Fault, southeast Tibet Plateau

Article Type: Research Paper

Keywords: Thermal spring; isotopes; hydrogeochemistry; earthquake

The purposes of the manuscript WATER-1507814 are to analyze the elemental composition and isotopic characteristics of thermal springs along the Weixi-Qiaohou Fault, southeast Tibet Plateau, and to explore the relationship between changes of the geochemical concentration in thermal spring water and the occurrence of earthquakes

The paper appears well-structured, however some sections must be improved.  Therefore I  believe

the manuscript should be published only after major revision.

Comments:

  1. Geological setting

Figure 1 is not a geological map! I suggest to add a new (and true) geological map!

  1. Data and methods

Nothing is said about the analytical precision and accuracy data!

Table 1 is too dispersive, I recommend moving it to supplementary materials

  1. Results

Only extreme values are reported in the discussion of the results! add at least the means and standard deviations

Move also table 4 in to supplementary materials

  1. Discussion

5.2. Hydrochemical characteristics of hot springs 180

5.2.1. Dissolved major elements

To evaluate the chemical composition of the water it is not enough to use the Piper diagram because it does not take into account (as proposed by the authors) salinity, I suggest using a TIS salinity diagram, as proposed by:

Apollaro, C.; Vespasiano, G.; De Rosa, R.: Marini L. Use of mean residence time and flowrate of thermal waters to evaluate the volume of reservoir water contributing to the natural discharge and the related geothermal reservoir volume. Application to Northern Thailand hot springs. Geothermics 2015, 58, 62-74

5.2.4. Reservoir temperature and circulation depth

It would be useful to add a table with the comparison between the cited geothermometers and temperatures provided from K-Na-Mg and Ca/Mg proposed by Chiodini et al., 1995.

5.3.2. Trace element origin

In the area outcrop a lot of different rocks, why do you have choose only the granitic rocks  for insert in the equation: EFi=(Ci/Ti)w/(Ci/Ti)r

ADD THESE REFERENCES

Apollaro, C.; Vespasiano, G.; De Rosa, R.: Marini L. Use of mean residence time and flowrate of thermal waters to evaluate the volume of reservoir water contributing to the natural discharge and the related geothermal reservoir volume. Application to Northern Thailand hot springs. Geothermics 2015, 58, 62-74

Chiodini, G., Frondini, F. and Marini, L. (1995). Theoretical geothermometers and  pCO2 indicators for aqueous solutions coming from hydrothermal systems of medium-low temperature hosted in carbonate-evaporite rocks. Application to the thermal springs of the Etruscan Swell. Italy. Applied Geochemistry, 10, 337-346.

Reviewer 3 Report

The paper “Hydrochemical characteristics of earthquake-related thermal springs along the Weixi-Qiaohou Fault, southeast Tibet Plateau” by Zhou et al. presents a complete dataset for the hydrogeochemical characterization of a series of thermal springs and report some hydrochemical anomalies possibly related with local seismicity. The paper well fits in the journal aims and the dataset seems consistent, however the interpretation of the results and the presentation of the paper hinder a possible higher impact of the observe results.

Starting from the introduction, authors should more critically focus on the known and the unknown issues related with hydrogeochemical responses to earthquakes, including also co- occurring and post-earthquake responses, in order to better highlight the complexity of these phenomena. I suggest the reading of some interesting and very recent research papers and reviews which try to combine all the mechanisms observed and worth to be mentioned in the introduction in my opinion: Martinelli et al., 2020 https://doi.org/10.3390/min11020107 ;  Binda et al., 2020 https://doi.org/10.3390/min10121058 ; Wang and Manga, 2021 https://doi.org/10.1007/978-3-030-64308-9_9 ; Chiodini et al., 2020 10.1126/sciadv.abc2938

Moreover, I suggest to the authors to better define the aims of the study and the following structure of the paper. As it is, the structure is a bit confounding: authors start to discuss the issues of major ions, trace elements and isotopes then mainly focus on major ions only for hydrogeochemical anomalies related to seismicity. Since the authors submitted this paper to a special issue concerning Earthquakes and Groundwater, I suggest to the authors to shorten the section discussing the general hydrochemistry of the springs and focus more on the effects of the earthquake sequence in these springs.

Then, in the methods, more details on the QA/QC protocols for samplings should be provided. E.g., use of blanks, material preparation, sample pre-treatments (filtration and acidification).

Moving to the results section, it can be joined with the discussion in my opinion. Another issue in the presentation of results is the labelling of samples. The double labelling of waters with the sampling site and the W numbering is confounding in my opinion. Authors should label the samples univocally.

There are different revisions needed for the set-up of figures and tables in my opinion. I think that table 1 can be moved to supplementary material. Authors should instead focus the discussion on the specific observed changes.

Then, table 2 and figure 2 are mostly redundant. Moreover, the interpretation of these data is generally concerning, since the period of the year is different in some samples, and the study area is known to have periodic precipitation events which can highly affect isotopic enrichment/depletion of water. This issue needs to be discussed also for other chemical variables, since seasonal trends can play as confounding factors for hydrogeochemical anomalies investigation. See e.g. Federico et al., 2004 https://doi.org/10.1016/S0377-0273(03)00392-5

The information reported in figure 4 is limited as it is in my opinion. Most of the interpretations of observed in lines 218-227 can be also reported from the interpretation of the piper diagram. Moreover, the difference among springs are not easy to observe. Authors should consider to add a panel showing the detail of the different springs, which mostly plot in the Mg corner.

Also, figure 5 is really hard to read. All the saturation indexes for all the springs in the same plot make the graph too messy in my opinion. Authors should better emphasize the SI showing changes among springs and the one which are instead stable, and then focus the discussion accordingly to understand which changes in mineral saturation can be related to seismicity.

Then, in Figure 8: measure uncertainties should be added to the graph. Moreover, to which springs are the graphs referring? This information is needed to define the hydrogeological issues discussed by the authors in the text.

The comparison of changes in major ions and the explanation of mechanisms in lines 351-360 and in lines 379 and 383 are a bit too simplistic in my opinion. Moving to the discussion, the comparison of changes in major ions and the explanation of mechanisms in lines 351-360 and in lines 379 and 383 are a bit too simplistic in my opinion. The mechanisms of responses can be widely different but showing similar changes, depending on the hydrogeological setting of the aquifer (see e.g. Rosen et al., 2018  https://doi.org/10.1029/2017WR022097 ). Authors should more critically compare their results with different hydrogeological settings of the references 48 and 77. Also, in figure 10, Observing the aspecific decrease of major ions in these springs, I would expect also a change in conductivity. Do the authors collected these data? It would be good to understand is these changes are related to a possible dilution of thermal water. This information can help to better identify the main mechanism for the anomalies.

Then, the main outlooks of the research are missing in the conclusion. Authors should add the expectable consequences of their work for the knowledge gaps in the field of detection of hydrogeochemical anomalies.

Finally, there are some minor issues which require revisions in my opinion:

-This information is not fundamental for the interpretation of the presented data and can be removed.

-lines 209-201: The meaning of the sentence is not clear. How do these springs not change? Do the authors mean in temporal trends or among the different springs? Please define it.

-lines 249-253: The definition of heat storage temperature and its calculation should be moved to the method section.

-Line 273: I think that tritium is a typo. It should be Strontium.

-Line 296: Authors should add a reference for this sentence.

-Line 300: Limestone is not a Li-containing silica.

-Line 427: Which is the reference for the altitude of 1.9-3.1 km? Is it below ground surface? Moreover, please add a scale bar in figure 11 to better highlight the depth and dimensions of the modeled area.

Round 2

Reviewer 2 Report

considering the revisions made, the work can be accepted

Author Response

Thank you for your comments!

Reviewer 3 Report

The revised version of the paper "Hydrochemical characteristics of earthquake-related thermal springs along the Weixi-Qiaohou Fault, southeast Tibet Plateau" has shown a clear improvement in clarity after the revisions. However, there are still several minor issues and spellchecks which require revisions:

-lines 59-60: Reference "Petrini et al" missing

-lines 63-65: please rephrase: "by measing different chemical variables: hydrogen and oxygen isotopes, major ions and trace elements".

-line 137: Revise the typo

-lines 143-145: please use commas instead of repeating "and"

-lines 157-160: It is not clear to which reproducibility the authors are talking about. I suggest to rephrase the sentence.

-line 237: "processes can not"

-line 366: "The instead of "Thw"

-line 396: SiO2 with lowercase

-line 399: please remove the extra comma

-line 451: please rephrase the sentence

-line 551: I think that " 61/4  " is a typo

-line 691: "analyzing" instead of "investigating"

Author Response

Thank you for your comments!

Round 3

Reviewer 3 Report

The revised version of the paper "Hydrochemical characteristics of earthquake-related thermal springs along the Weixi-Qiaohou Fault, southeast Tibet Plateau" can be now published in my opinion.

Author Response

Thank you for your comments